# Banach Wasserstein GAN

**Jonas Adler**
Department of Mathematics
KTH - Royal institute of Technology
Research and Physics
Elekta
jonasadl@kth.se

**Sebastian Lunz**
Department of Applied Mathematics
and Theoretical Physics
University of Cambridge
lunz@math.cam.ac.uk

## Abstract

Wasserstein Generative Adversarial Networks (WGANs) can be used to generate realistic samples from complicated image distributions. The Wasserstein metric used in WGANs is based on a notion of distance between individual images, which induces a notion of distance between probability distributions of images. So far the community has considered $\ell^2$ as the underlying distance. We generalize the theory of WGAN with gradient penalty to Banach spaces, allowing practitioners to select the features to emphasize in the generator. We further discuss the effect of some particular choices of underlying norms, focusing on Sobolev norms. Finally, we demonstrate a boost in performance for an appropriate choice of norm on CIFAR-10 and CelebA.

## 1 Introduction

Generative Adversarial Networks (GANs) are one of the most popular generative models [6]. A neural network, the *generator*, learns a map that takes random input noise to samples from a given distribution. The training involves using a second neural network, the *critic*, to discriminate between real samples and the generator output.

In particular, [2, 7] introduces a critic built around the Wasserstein distance between the distribution of true images and generated images. The Wasserstein distance is inherently based on a notion of distance between images which in all implementations of Wasserstein GANs (WGAN) so far has been the $\ell^2$ distance. On the other hand, the imaging literature contains a wide range of metrics used to compare images [4] that each emphasize different features of interest, such as edges or to more accurately approximate human observer perception of the generated image.

There is hence an untapped potential in selecting a norm beyond simply the classical $\ell^2$ norm. We could for example select an appropriate Sobolev space to either emphasize edges, or large scale behavior. In this work we extend the classical WGAN theory to work on these and more general Banach spaces.

Our contributions are as follows:

- We introduce Banach Wasserstein GAN (BWGAN), extending WGAN implemented via a gradient penalty (GP) term to any separable complete normed space.
- We describe how BWGAN can be efficiently implemented. The only practical difference from classical WGAN with gradient penalty is that the $\ell^2$ norm is replaced with a dual norm. We also give theoretically grounded heuristics for the choice of regularization parameters.
- We compare BWGAN with different norms on the CIFAR-10 and CelebA datasets. Using the Space $L^{10}$, which puts strong emphasize on outliers, we achieve an unsupervised inception score of $8.31$ on CIFAR-10, state of the art for non-progressive growing GANs.

## 2 Background

### 2.1 Generative adversarial networks

Generative Adversarial Networks (GANs) [6] perform generative modeling by learning a map $G : Z \to B$ from a low-dimensional latent space $Z$ to image space $B$, mapping a fixed noise distribution $\mathbb{P}_Z$ to a distribution of generated images $\mathbb{P}_G$.

In order to train the generative model $G$, a second network $D$ is used to discriminate between original images drawn from a distribution of real images $\mathbb{P}_r$ and images drawn from $\mathbb{P}_G$. The generator is trained to output images that are conceived to be realistic by the critic $D$. The process is iterated, leading to the famous minimax game [6] between generator $G$ and critic $D$

$$\min_G \max_D \mathbb{E}_{X \sim \mathbb{P}_r} \left[ \log(D(X)) \right] + \mathbb{E}_{Z \sim \mathbb{P}_Z} \left[ \log(1 - D(G_\Theta(Z))) \right]. \tag{1}$$

Assuming the discriminator is perfectly trained, this gives rise to the Jensen–Shannon divergence (JSD) as distance measure between the distributions $\mathbb{P}_G$ and $\mathbb{P}_r$ [6, Theorem 1].

### 2.2 Wasserstein metrics

To overcome undesirable behavior of the JSD in the presence of singular measures [1], in [2] the Wasserstein metric is introduced to quantify the distance between the distributions $\mathbb{P}_G$ and $\mathbb{P}_r$. While the JSD is a strong metric, measuring distances point-wise, the Wasserstein distance is a weak metric, measuring the cost of transporting one probability distribution to another. This allows it to stay finite and provide meaningful gradients to the generator even when the measures are mutually singular.

In a rather general form, the Wasserstein metric takes into account an underlying metric $d_B : B \times B \to \mathbb{R}$ on a Polish (e.g. separable completely metrizable) space $B$. In its primal formulation, the Wasserstein-$p$, $p \geq 1$, distance is defined as

$$\text{Wass}_p(\mathbb{P}_G, \mathbb{P}_r) := \left( \inf_{\pi \in \Pi(\mathbb{P}_G, \mathbb{P}_r)} \mathbb{E}_{(X_1, X_2) \sim \pi} d_B(X_1, X_2)^p \right)^{1/p}, \tag{2}$$

where $\Pi(\mathbb{P}_G, \mathbb{P}_r)$ denotes the set of distributions on $B \times B$ with marginals $\mathbb{P}_G$ and $\mathbb{P}_r$. The Wasserstein distance is hence highly dependent on the choice of metric $d_B$.

The Kantorovich-Rubinstein duality [19, 5.10] provides a way of more efficiently computing the Wasserstein-1 distance (which we will henceforth simply call the Wasserstein distance, $\text{Wass} = \text{Wass}_1$) between measures on high dimensional spaces. The duality holds in the general setting of Polish spaces and states that

$$\text{Wass}(\mathbb{P}_G, \mathbb{P}_r) = \sup_{\text{Lip}(f) \leq 1} \mathbb{E}_{X \sim \mathbb{P}_G} f(X) - \mathbb{E}_{X \sim \mathbb{P}_r} f(X). \tag{3}$$

The supremum is taken over all Lipschitz continuous functions $f : B \to \mathbb{R}$ with Lipschitz constant equal or less than one. We note that in this dual formulation, the dependence of $f$ on the choice of metric is encoded in the condition of $f$ being 1-Lipschitz and recall that a function $f : B \to \mathbb{R}$ is $\gamma$-Lipschitz if

$$|f(x) - f(y)| \leq \gamma d_B(x, y).$$

In an abstract sense, the Wasserstein metric could be used in GAN training by using a critic $D$ to approximate the supremum in (3). The generator uses the loss $\mathbb{E}_{Z \sim \mathbb{P}_Z} D(G(Z))$. In the case of a perfectly trained critic $D$, this is equivalent to using the Wasserstein loss $\text{Wass}(\mathbb{P}_G, \mathbb{P}_r)$ to train $G$ [2, Theorem 3].

### 2.3 Wasserstein GAN

Implementing GANs with the Wasserstein metric requires to approximate the supremum in (3) with a neural network. In order to do so, the Lipschitz constraint has to be enforced on the network. In the paper Wasserstein GAN [2] this was achieved by restricting all network parameters to lie within a predefined interval. This technique typically guarantees that the network is $\gamma$ Lipschitz for some $\gamma$ for *any* metric space. However, it typically reduces the set of admissible functions to a proper subset

of all $\gamma$ Lipschitz functions, hence introducing an uncontrollable additional constraint on the network. This can lead to training instabilities and artifacts in practice [7].

In [7] strong evidence was presented that the condition can better be enforced by working with another characterization of $1-$Lipschitz functions. In particular, they prove that if $B = \mathbb{R}^n, d(x,y)_B = \|x-y\|_2$ we have the gradient characterization

$$f \text{ is } 1 - \text{Lipschitz} \iff \|\nabla f(x)\|_2 \leq 1 \quad \text{for all } x \in \mathbb{R}^n.$$

They softly enforce this condition by adding a penalty term to the loss function of $D$ that takes the form

$$\mathbb{E}_{\hat{X}} \left( \|\nabla D(\hat{X})\|_2 - 1 \right)^2, \tag{4}$$

where the distribution of $\hat{X}$ is taken to be the uniform distributions on lines connecting points drawn from $\mathbb{P}_G$ and $\mathbb{P}_r$.

However, penalizing the $\ell^2$ norm of the gradient corresponds specifically to choosing the $\ell^2$ norm as underlying distance measure on image space. Some research has been done on generalizing GAN theory to other spaces [18, 11], but in its current form WGAN with gradient penalty does not extend to arbitrary choices of underlying spaces $B$. We shall give a generalization to a large class of spaces, the (separable) Banach spaces, but first we must introduce some notation.

## 2.4 Banach spaces of images

A vector space is a collection of objects (vectors) that can be added together and scaled, and can be seen as a generalization of the Euclidean space $\mathbb{R}^n$. If a vector space $B$ is equipped with a notion of length, a norm $\|\cdot\|_B : B \to \mathbb{R}$, we call it a normed space. The most commonly used norm is the $\ell^2$ norm defined on $\mathbb{R}^n$, given by

$$\|x\|_2 = \left( \sum_{i=1}^n x_i^2 \right)^{1/2}.$$

Such spaces can be used to model images in a very general fashion. In a pixelized model, the image space $B$ is given by the discrete pixel values, $B \sim \mathbb{R}^{n \times n}$. Continuous image models that do not rely on the concept of pixel discretization include the space of square integrable functions over the unit square. The norm $\|\cdot\|_B$ gives room for a choice on how distances between images are measured. The Euclidean distance is a common choice, but many other distance notions are possible that account for more specific image features, like the position of edges in Sobolev norms.

A normed space is called a Banach space if it is complete, that is, Cauchy sequences converge. Finally, a space is separable if there exists some countable dense subset. Completeness is required in order to ensure that the space is rich enough for us to define limits whereas separability is necessary for the usual notions of probability to hold. These technical requirements formally hold in finite dimensions but are needed in the infinite dimensional setting. We note that all separable Banach spaces are Polish spaces and we can hence define Wasserstein metrics on them using the induced metric $d_B(x,y) = \|x-y\|_B$.

For any Banach space $B$, we can consider the space of all bounded linear functionals $B \to \mathbb{R}$, which we will denote $B^*$ and call the (topological) dual of $B$. It can be shown [17] that this space is itself a Banach space with norm $\|\cdot\|_{B^*} : B^* \to \mathbb{R}$ given by

$$\|x^*\|_{B^*} = \sup_{x \in B} \frac{x^*(x)}{\|x\|_B}. \tag{5}$$

In what follows, we will give some examples of Banach spaces along with explicit characterizations of their duals. We will give the characterizations in continuum, but they are also Banach spaces in their discretized (finite dimensional) forms.

$L^p$**-spaces.** Let $\Omega$ be some domain, for example $\Omega = [0,1]^2$ to model square images. The set of functions $x : \Omega \to \mathbb{R}$ with norm

$$\|x\|_{L^p} = \left( \int_\Omega x(t)^p dt \right)^{1/p} \tag{6}$$

is a Banach space with dual $[L^p]^* = L^q$ where $1/p + 1/q = 1$. In particular, we note that $[L^2]^* = L^2$. The parameter $p$ controls the emphasis on outliers, with higher values corresponding to a stronger focus on outliers. In the extreme case $p = 1$, the norm is known to induce sparsity, ensuring that all but a small amount of pixels are set to the correct values.

**Sobolev spaces.** Let $\Omega$ be some domain, then the set of functions $x : \Omega \to \mathbb{R}$ with norm

$$\|x\|_{W^{1,2}} = \left( \int_\Omega x(t)^2 + |\nabla x(t)|^2 dt \right)^{1/2}$$

where $\nabla x$ is the spatial gradient, is an example of a *Sobolev* space. In this space, more emphasis is put on the edges than in e.g. $L^p$ spaces, since if $\|x_1 - x_2\|_{W^{1,2}}$ is small then not only are their absolute values close, but so are their edges.

Since taking the gradient is equivalent to multiplying with $\xi$ in the Fourier space, the concept of Sobolev spaces can be generalized to arbitrary (real) derivative orders $s$ if we use the norm

$$\|x\|_{W^{s,p}} = \left( \int_\Omega \left( \mathcal{F}^{-1} \left[ (1 + |\xi|^2)^{s/2} \mathcal{F} x \right] (t) \right)^p dt \right)^{1/p}, \tag{7}$$

where $\mathcal{F}$ is the Fourier transform. The tuning parameter $s$ allows to control which frequencies of an image are emphasized: A negative value of $s$ corresponds to amplifying low frequencies, hence prioritizing the global structure of the image. On the other hand, high values of $s$ amplify high frequencies, thus putting emphasis on sharp local structures, like the edges or ridges of an image.

The dual of the Sobolev space, $[W^{s,p}]^*$, is $W^{-s,q}$ where $q$ is as above [3]. Under weak assumptions on $\Omega$, all Sobolev spaces with $1 \le p < \infty$ are separable. We note that $W^{0,p} = L^p$ and in particular we recover as an important special case $W^{0,2} = L^2$.

There is a wide range of other norms that can be defined for images, see appendix A and [5, 3] for a further overview of norms and their respective duals.

## 3 Banach Wasserstein GANs

In this section we generalize the loss (4) to separable Banach spaces, allowing us to effectively train a Wasserstein GAN using arbitrary norms.

We will show that the characterization of $\gamma$-Lipschitz functions via the norm of the differential can be extended from the $\ell_2$ setting in (4) to arbitrary Banach spaces by considering the gradient as an element in the dual of $B$. In particular, for any Banach space $B$ with norm $\|\cdot\|_B$, we will derive the loss function

$$L = \frac{1}{\gamma} \left( \mathbb{E}_{X \sim \mathbb{P}_\Theta} D(X) - \mathbb{E}_{X \sim \mathbb{P}_r} D(X) \right) + \lambda \mathbb{E}_{\hat{X}} \left( \frac{1}{\gamma} \|\partial D(\hat{X})\|_{B^*} - 1 \right)^2, \tag{8}$$

where $\lambda, \gamma \in \mathbb{R}$ are regularization parameters, and show that a minimizer of this this is an approximation to the Wasserstein distance on $B$.

### 3.1 Enforcing the Lipschitz constraint in Banach spaces

Throughout this chapter, let $B$ denote a Banach space with norm $\|\cdot\|_B$ and $f : B \to \mathbb{R}$ a continuous function. We require a more general notion of gradient: The function $f$ is called Fréchet differentiable at $x \in B$ if there is a bounded linear map $\partial f(x) : B \to \mathbb{R}$ such that

$$\lim_{\|h\|_B \to 0} \frac{1}{\|h\|_B} \left| f(x + h) - f(x) - \left[ \partial f(x) \right](h) \right| = 0. \tag{9}$$

The differential $\partial f(x)$ is hence an element of the dual space $B^*$. We note that the usual notion of gradient $\nabla f(x)$ in $\mathbb{R}^n$ with the standard inner product is connected to the Fréchet derivative via $\left[ \partial f(x) \right](h) = \nabla f(x) \cdot h$.

The following theorem allows us to characterize all Lipschitz continuous functions according to the dual norm of the Fréchet derivative.

**Lemma 1.** *Assume $f : B \to \mathbb{R}$ is Fréchet differentiable. Then $f$ is $\gamma$-Lipschitz if and only if*

$$\|\partial f(x)\|_{B^*} \leq \gamma \quad \forall x \in B. \tag{10}$$

*Proof.* Assume $f$ is $\gamma$-Lipschitz. Then for all $x, h \in B$ and $\epsilon > 0$

$$\left[\partial f(x)\right](h) = \lim_{\epsilon \to 0} \frac{1}{\epsilon}(f(x + \epsilon h) - f(x)) \leq \lim_{\epsilon \to 0} \frac{\gamma\epsilon\|h\|_B}{\epsilon} = \gamma\|h\|_B,$$

hence by the definition of the dual norm, eq. (5), we have

$$\|\partial f(x)\|_{B^*} = \sup_{h \in B} \frac{\left[\partial f(x)\right](h)}{\|h\|_B} \leq \sup_{h \in B} \frac{\gamma\|h\|_B}{\|h\|_B} \leq \gamma.$$

Now let $f$ satisfy (10) and let $x, y \in B$. Define the function $g : \mathbb{R} \to \mathbb{R}$ by

$$g(t) = f(x(t)), \qquad \text{where} \quad x(t) = tx + (1 - t)y,$$

As $x(t + \Delta t) - x(t) = \Delta t(x - y)$, we see that $g$ is everywhere differentiable and

$$g'(t) = \left[\partial f\big(x(t)\big)\right](x - y).$$

Hence

$$|g'(t)| = \left|\left[\partial f\big(x(t)\big)\right](x - y)\right| \leq \|\partial f(x(t))\|_{B^*}\|x - y\|_B \leq \gamma\|x - y\|_B,$$

which gives

$$|f(x) - f(y)| = |g(1) - g(0)| \leq \int_0^1 |g'(t)| \, dt \leq \gamma\|x - y\|_B,$$

thus finishing the proof. $\qquad\qquad\qquad\qquad\qquad\qquad\qquad\qquad\qquad\qquad\qquad\qquad\qquad\square$

Using lemma 1 we see that a $\gamma$-Lipschitz requirement in Banach spaces is equivalent to the dual norm of the Fréchet derivative being less than $\gamma$ everywhere. In order to enforce this we need to compute $\|\partial f(x)\|_{B^*}$. As shown in section 2.4, the dual norm can be readily computed for a range of interesting Banach spaces, but we also need to compute $\partial f(x)$, preferably using readily available automatic differentiation software. However, such software can typically only compute derivatives in $\mathbb{R}^n$ with the standard norm.

Consider a finite dimensional Banach space $B$ equipped by *any* norm $\|\cdot\|_B$. By Lemma 1, gradient norm penalization requires characterizing (e.g. giving a basis for) the dual $B^*$ of $B$. This can be a difficult for infinite dimensional Banach spaces. In a finite dimensional however setting, there is an linear continuous bijection $\iota : \mathbb{R}^n \to B$ given by

$$\iota(x)_i = x_i. \tag{11}$$

This isomorphism implicitly relies on the fact that a basis of $B$ can be chosen and can be mapped to the corresponding dual basis. This does not generalize to the infinite dimensional setting, but we hope that this is not a very limiting assumption in practice.

We note that we can write $f = g \circ \iota$ where $g : \mathbb{R}^n \to \mathbb{R}$ and automatic differentiation can be used to compute the derivative $\partial g(x)$ efficiently. Further, note that the chain rule yields

$$\partial f(x) = \iota^* \left(\partial g(\iota(x))\right),$$

where $\iota^* : \mathbb{R}^n \to B^*$ is the adjoint of $\iota$ which is readily shown to be as simple as $\iota$, $\iota^*(x)_i = x_i$. This shows that computing derivatives in finite dimensional Banach spaces can be done using standard automatic differentiation libraries with only some formal mathematical corrections. In an implementation, the operators $\iota, \iota^*$ would be implicit.

In terms of computational costs, the difference between general Banach Wasserstein GANs and the ones based on the $\ell^2$ metric lies in the computation of the gradient of the dual norm. By the chain rule, any computational step outside the calculation of this gradient is the same for any choice of underlying notion of distance. This in particular includes any forward pass or backpropagation step through the layers of the network used as discriminator. If there is an efficient framework available to compute the gradient of the dual norm, as in the case of the Fourier transform used for Sobolev spaces, the computational expenses hence stay essentially the same independent of the choice of norm.

## 3.2 Regularization parameter choices

The network will be trained by adding the regularization term

$$\lambda \mathbb{E}_{\hat{X}} \left( \frac{1}{\gamma} \|\partial D(\hat{X})\|_{B^*} - 1 \right)^2.$$

Here, $\lambda$ is a regularization constant and $\gamma$ is a scaling factor controlling which norm we compute. In particular $D$ will approximate $\gamma$ times the Wasserstein distance. In the original WGAN-GP paper [7] and most following work $\lambda = 10$ and $\gamma = 1$, while $\gamma = 750$ was used in Progressive GAN [9]. However, it is easy to see that these values are specific to the $\ell_2$ norm and that we would need to re-tune them if we change the norm. In order to avoid having to hand-tune these for every choice of norm, we will derive some heuristic parameter choice rules that work with any norm.

For our heuristic, we will start by assuming that the generator is the zero-generator, always returning zero. Assuming symmetry of the distribution of true images $\mathbb{P}_r$, the discriminator will then essentially be decided by a single constant $f(x) = c\|x\|_B$, where $c$ solves the optimization problem

$$\min_{c \in \mathbb{R}} \mathbb{E}_{X \sim \mathbb{P}_r} \left[ -\frac{c\|X\|_B}{\gamma} + \frac{\lambda(c - \gamma)^2}{\gamma^2} \right].$$

By solving this explicitly we find

$$c = \gamma \left( 1 + \frac{\mathbb{E}_{X \sim \mathbb{P}_r} \|X\|_B}{2\lambda} \right).$$

Since we are trying to approximate $\gamma$ times the Wasserstein distance, and since the norm has Lipschitz constant 1, we want $c \approx \gamma$. Hence to get a small relative error we need $\mathbb{E}_{X \sim \mathbb{P}_r} \|X\|_B \ll 2\lambda$. With this theory to guide us, we can make the heuristic rule

$$\lambda \approx \mathbb{E}_{X \sim \mathbb{P}_r} \|X\|_B.$$

In the special case of CIFAR-10 with the $\ell_2$ norm this gives $\lambda \approx 27$, which agrees with earlier practice ($\lambda = 10$) reasonably well.

Further, in order to keep the training stable we assume that the network should be approximately scale preserving. Since the operation $x \rightarrow \partial D(x)$ is the deepest part of the network (twice the depth as the forward evaluation), we will enforce $\|x\|_{B^*} \approx \|\partial D(x)\|_{B^*}$. Assuming $\lambda$ was appropriately chosen, we find in general (by lemma 1) $\|\partial D(x)\|_{B^*} \approx \gamma$. Hence we want $\gamma \approx \|x\|_{B^*}$. We pick the expected value as a representative and hence we obtain the heuristic

$$\gamma \approx \mathbb{E}_{X \sim \mathbb{P}_r} \|X\|_{B^*}$$

For CIFAR-10 with the $\ell_2$ norm this gives $\gamma = \lambda \approx 27$ and may explain the improved performance obtained in [9].

A nice property of the above parameter choice rules is that they can be used with any underlying norm. By using these parameter choice rules we avoid the issue of hand tuning further parameters when training using different norms.

## 4 Computational results

To demonstrate computational feasibility and to show how the choice of norm can impact the trained generator, we implemented Banach Wasserstein GAN with various Sobolev and $L^p$ norms, applied to CIFAR-10 and CelebA ($64 \times 64$ pixels). The implementation was done in TensorFlow and the architecture used was a faithful re-implementation of the residual architecture used in [7], see appendix B. For the loss function, we used the loss eq. (8) with parameters according to section 3.2 and the norm chosen according to either the Sobolev norm eq. (7) or the $L^p$ norm eq. (6). In the case of the Sobolev norm, we selected units such that $|\xi| \leq 5$. Following [9], we add a small $10^{-5} \mathbb{E}_{X \sim \mathbb{P}_r} D(X)^2$ term to the discriminator loss to stop it from drifting during the training.

For training we used the Adam optimizer [10] with learning rate decaying linearly from $2 \cdot 10^{-4}$ to 0 over $100\,000$ iterations with $\beta_1 = 0$, $\beta_2 = 0.9$. We used 5 discriminator updates per generator update. The batch size used was 64. In order to evaluate the reproducibility of the results on CIFAR-10, we

Figure 1: Generated CIFAR-10 samples for some $L^p$ spaces.

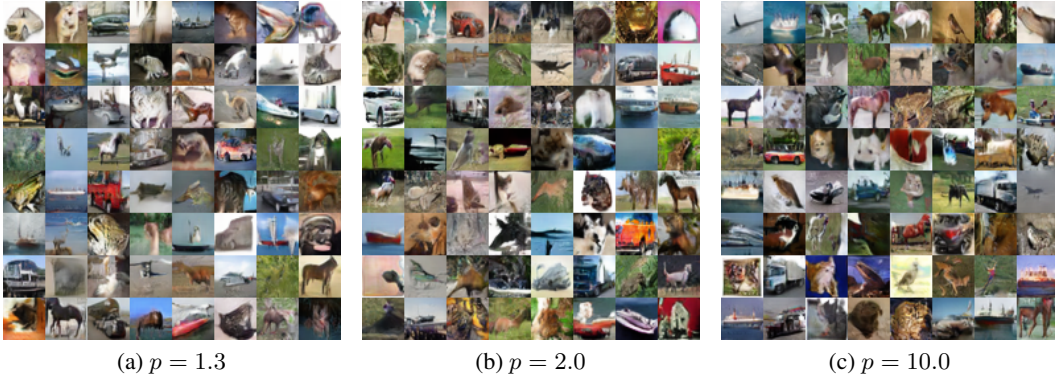

(a) $p = 1.3$          (b) $p = 2.0$          (c) $p = 10.0$

Figure 2: FID scores for BWGAN on CIFAR-10.

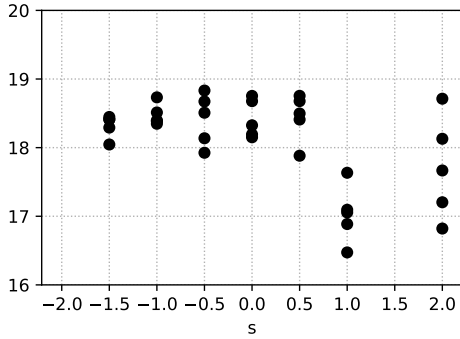
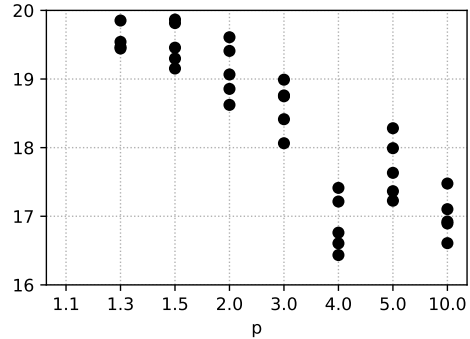

(a) $W^{s,2}$          (b) $L^p$

Figure 3: Inception scores for BWGAN on CIFAR-10.    Figure 4: Inception Scores on CIFAR-10.

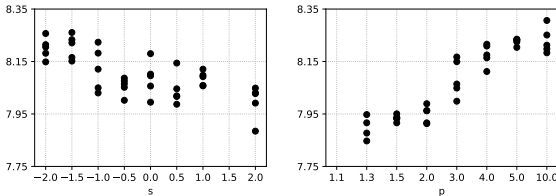

| Method | Inception Score |
|---|---|
| DCGAN [16] | $6.16 \pm .07$ |
| EBGAN [21] | $7.07 \pm .10$ |
| WGAN-GP [7] | $7.86 \pm .07$ |
| CT GAN [20] | $8.12 \pm .12$ |
| SNGAN [14] | $8.22 \pm .05$ |
| $W^{-\frac{3}{2},2}$-BWGAN | $8.26 \pm .07$ |
| $L^{10}$-BWGAN | $8.31 \pm .07$ |
| Progressive GAN [9] | $8.80 \pm .05$ |

followed this up by training an ensemble of 5 generators using SGD with warm restarts following [12]. Each warm restart used 10 000 generator steps. Our implementation is available online[1].

Some representative samples from the generator on both datasets can be seen in figs. 1 and 5. See appendix C for samples from each of the $W^{s,2}$ and $L^p$ spaces investigated as well as samples from the corresponding Fréchet derivatives.

For evaluation, we report Fréchet Inception Distance (FID)[8] and Inception scores, both computed from 50K samples. A high image quality corresponds to high Inception and low FID scores. On the CIFAR-10 dataset, both FID and inception scores indicate that negative $s$ and large values of $p$ lead to better image quality. On CelebA, the best FID scores are obtained for values of $s$ between $-1$ and $0$ and around $p = 0$, whereas the training become unstable for $p = 10$. We further compare our CIFAR-10 results in terms of Inception scores to existing methods, see table 4. To the best of

Figure 5: Generated CelebA samples for Sobolev spaces $W^{s,2}$.

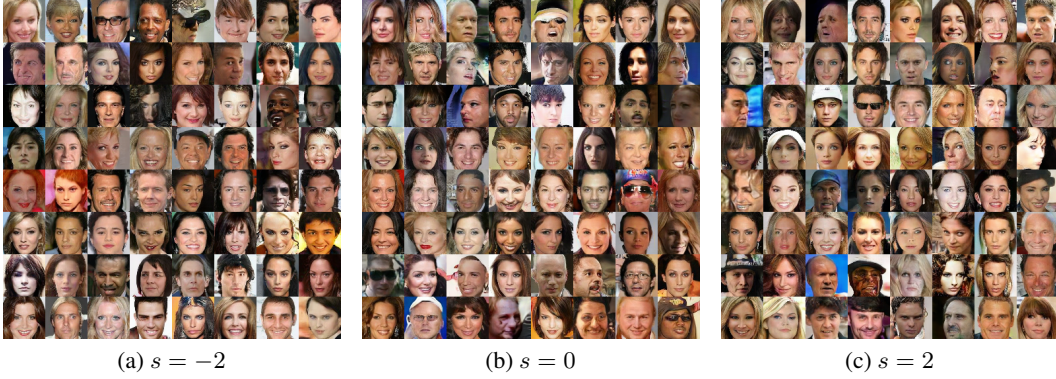

(a) $s = -2$          (b) $s = 0$          (c) $s = 2$

Figure 6: FID scores for BWGAN on CelebA.

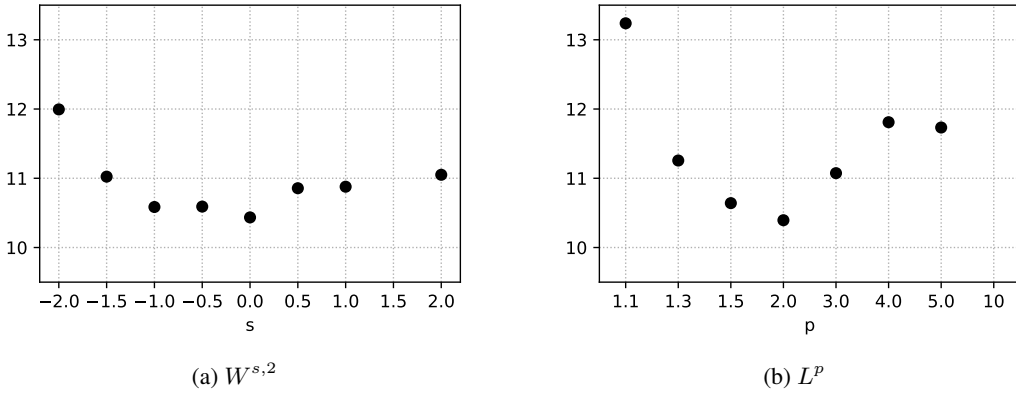

(a) $W^{s,2}$                (b) $L^p$

our knowledge, the inception score of $8.31 \pm 0.07$, achieved using the $L^{10}$ space, is state of the art for non-progressive growing methods. Our FID scores are also highly competitive, for CIFAR-10 we achieve $16.43$ using $L^4$. We also note that our result for $W^{0,2} = \ell^2$ is slightly better than the reference implementation, despite using the same network. We suspect that this is due to our improved parameter choices.

## 5 How about metric spaces?

Gradient norm penalization according to lemma 1 is only valid in Banach spaces, but a natural alternative to penalizing gradient norms is to enforce the Lipschitz condition directly (see [15]). This would potentially allow training Wasserstein GAN on general metric spaces by adding a penalty term of the form

$$\mathbb{E}_{X,Y} \left[ \left( \frac{|f(X) - f(Y)|}{d_B(X,Y)} - 1 \right)_+^2 \right]. \tag{12}$$

While theoretically equivalent to gradient norm penalization when the distributions of $X$ and $Y$ are chosen appropriately, this term is very likely to have considerably higher variance in practice.

For example, if we assume that $d$ is not bounded from below and consider two points $x, y \in M$ that are sufficiently close then a penalty term of the Lipschitz quotient as in (12) imposes a condition on the differential around $x$ and $y$ in the direction $(x - y)$ only, i.e. only $|\partial f(\tilde{x})(x - y)| \leq 1$ is ensured. In the case of two distributions that are already close, we will with high probability sample the difference quotient in a spatial direction that is parallel to the data, hence not exhausting the Lipschitz constraint, i.e. $|\partial f(\tilde{x})(x - y)| \ll 1$. Difference quotient penalization (12) then does not effectively enforce the Lipschitz condition. Gradient norm penalization on the other hand ensures this condition in all spatial directions simultaneously by considering the dual norm of the differential.

On the other hand, if $d$ is bounded from below the above argument fails. For example, Wasserstein GAN over a space equipped with the trivial metric

$$d_{\text{trivial}}(x, y) = \begin{cases} 0 & \text{if } x = y \\ 1 & \text{else} \end{cases}$$

approximates the Total Variation distance [19]. Using the regularizer eq. (12) we get a slight variation of Least Squares GAN [13]. We do not further investigate this line of reasoning.

## 6   Conclusion

We analyzed the dependence of Wasserstein GANs (WGANs) on the notion of distance between images and showed how choosing distances other than the $\ell^2$ metric can be used to make WGANs focus on particular image features of interest. We introduced a generalization of WGANs with gradient norm penalization to Banach spaces, allowing to easily implement WGANs for a wide range of underlying norms on images. This opens up a new degree of freedom to design the algorithm to account for the image features relevant in a specific application.

On the CIFAR-10 and CelebA dataset, we demonstrated the impact a change in norm has on model performance. In particular, we computed FID scores for Banach Wasserstein GANs using different Sobolev spaces $W^{s,p}$ and found a correlation between the values of both $s$ and $p$ with model performance.

While this work was motivated by images, the theory is general and can be applied to data in any normed space. In the future, we hope that practitioners take a step back and ask themselves if the $\ell^2$ metric is really the best measure of fit, or if some other metric better emphasize what they want to achieve with their generative model.

#### Acknowledgments

The authors would like to acknowledge Peter Maass for brining us together as well as important support from Ozan Öktem, Axel Ringh, Johan Karlsson, Jens Sjölund, Sam Power and Carola Schönlieb.

The work by J.A. was supported by the Swedish Foundation of Strategic Research grants AM13-0049, ID14-0055 and Elekta. The work by S.L. was supported by the EPSRC grant EP/L016516/1 for the University of Cambridge Centre for Doctoral Training, and the Cambridge Centre for Analysis. We also acknowledge the support of the Cantab Capital Institute for the Mathematics of Information.

## Footnotes

[1] `https://github.com/adler-j/bwgan`

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
