[Supplementary Material]

# A  Some further Banach spaces

There is some algebra for how to form new Banach spaces from known spaces. Specifically we have the following constructions that the reader might find useful.

**Weighted spaces.**  Let $B_1$ be some separable Banach space with norm $\|\cdot\|_{B_1}$, then we can construct another space $B_2$ with norm

$$\|f\|_{B_2} := \|Af\|_{B_1}$$

where $A : B_2 \to B_1$ is a continuous linear bijection. It is straightforward to show that the dual space $B_2^*$ has norm

$$\|f^*\|_{B_2^*} = \|A^{-*}f^*\|_{B_1^*}$$

where $A^{-*} : B_2^* \to B_1^*$ is the adjoint of the inverse of $A$. These weighted spaces could be used to focus on some feature of interest, e.g. focus especially on the red color channel or on some spatial region of the image, the center perhaps, that is more important.

**Product spaces.**  Let $B_1, \ldots B_n$ be Banach spaces and let $B = B_1 \times \cdots \times B_n$ be the product space with norm

$$\|(x_1, \ldots, x_n)\|_B = \left( \sum_{i=1}^{n} \|x_i\|_{B_i}^p \right)^{1/p}$$

then the dual space has norm

$$\|(x_1^*, \ldots, x_n^*)\|_{B^*} = \left( \sum_{i=1}^{n} \|x_i^*\|_{B_i^*}^q \right)^{1/q}$$

where $1/p + 1/q = 1$. These spaces could be used to explicitly model the color channels or even to model multi-modal data such as a generator outputting both an image and a caption.

# B  Network details

The implementation on CIFAR-10 faithfully follows the source code from [7]. It uses of residual blocks consisting of "nonlinearity + conv + nonlinearity + conv + residual connection" and meanpooling/nearest neighbor interpolation as building blocks. The generator starts from a latent space of 128 normally distributed random numbers and applies a dense layer to 4x4 images and applies a residual block then an interpolation repeatedly until the resolution 32x32 is reached. Then, a nonlinearity followed by a 1x1 convolution with 3 output channels is applied in order to obtain the generated color images.

The discriminator goes the other way using pooling with a final spatial mean-pooling followed by a dense layer. We used ReLU nonlinearities, all convolutions uses 128 channels and we used batch normalization after the nonlinearities in the generator. Following, we used uniform He initialization for all convolutions except the residual connections which used uniform Xavier initialization.

The implementation for CelebA follows that for CIFAR-10, with an additional residual block for further up/downsampling added both in the generator and discriminator.

See [7] and/or our open source implementation for further details.

# C  Further samples

We give samples from each of the Sobolev spaces $W^{s,2}$ investigated in the paper. The qualitative results mirror those observed in section 4 with higher $s$ indicating higher gradients in the discriminators Fréchet derivative, thus indicating a focus on higher frequency content. We also show further examples along with the corresponding loss gradients for some $L^p$ spaces on both the CelebA and CIFAR-10 dataset.

(a) $s = -2$

(b) $s = 0.0$

(c) $s = -\frac{3}{2}$

(d) $s = 0.5$

(e) $s = -1.0$

(f) $s = 1.0$

(g) $s = -0.5$

(h) $s = 2.0$

Figure 7: Samples for all $W^{s,2}$-spaces investigated on CIFAR-10.

(a) $s = -2$

(b) $s = 0.0$

(c) $s = -\frac{3}{2}$

(d) $s = 0.5$

(e) $s = -1.0$

(f) $s = 1.0$

(g) $s = -0.5$

(h) $s = 2.0$

Figure 8: Fréchet derivatives for all $W^{s,2}$-spaces investigated on CIFAR-10.

(a) $s = -2$

(b) $s = 0.0$

(c) $s = -\frac{3}{2}$

(d) $s = 0.5$

(e) $s = -1.0$

(f) $s = 1.0$

(g) $s = -0.5$

(h) $s = 2.0$

Figure 9: Samples for all $W^{s,2}$-spaces investigated on CelebA.

(a) $s = -2$

(b) $s = 0.0$

(c) $s = -\frac{3}{2}$

(d) $s = 0.5$

(e) $s = -1.0$

(f) $s = 1.0$

(g) $s = -0.5$

(h) $s = 2.0$

Figure 10: Fréchet derivatives for all $W^{s,2}$-spaces investigated on CelebA.

(a) $p = 1.1$

(b) $p = 3.0$

(c) $p = 1.3$

(d) $p = 4.0$

(e) $p = 1.5$

(f) $p = 5.0$

(g) $p = 2.0$

(h) $p = 10.0$

Figure 11: Samples from all $L^p$-spaces investigated on CIFAR-10.

(a) $p = 1.1$

(b) $p = 3.0$

(c) $p = 1.3$

(d) $p = 4.0$

(e) $p = 1.5$

(f) $p = 5.0$

(g) $p = 2.0$

(h) $p = 10.0$

Figure 12: Fréchet derivatives for all $L^p$-spaces investigated on CIFAR-10.

(a) $p = 1.1$

(b) $p = 3.0$

(c) $p = 1.3$

(d) $p = 4.0$

(e) $p = 1.5$

(f) $p = 5.0$

(g) $p = 2.0$

Failed to train

(h) $p = 10.0$

Figure 13: Samples from all $L^p$-spaces investigated on CelebA.

(a) $p = 1.1$

(b) $p = 3.0$

(c) $p = 1.3$

(d) $p = 4.0$

(e) $p = 1.5$

(f) $p = 5.0$

(g) $p = 2.0$

Failed to train
(h) $p = 10.0$

Figure 14: Fréchet derivatives for all $L^p$-spaces investigated on CelebA.