[Reviews · NeurIPS 2018]

Reviewer 1



Update: After reading the authors' response, my opinion remains that this is a strong paper and I argue for its acceptance. The addition of FID and emphasis of the L^p norms further improve it. I do still suggest that the authors run experiments on at least one other dataset: I agree that more datasets would "contribute relatively little to the reader’s understanding of the proposed method", but they do serve a purpose of convincing readers of the practical effectiveness and versatility of that method. I also don't think the authors need to present the same number of results for each dataset; one experiment per dataset demonstrating improvement seems sufficient. --- The popular WGAN-GP method of GAN training minimizes a loss which is defined in terms of an L^2 ground metric. This paper generalizes that method to arbitrary Banach spaces and in particular explores training WGAN-GPs with Sobolev norms, which are perhaps a more sensible choice for image modeling. Experimentally, the proposed BWGAN achieves very good Inception scores on CIFAR-10. This work directly addresses an important but under-explored aspect of WGANs, namely the importance of the ground metric. The choice of L^2 was made originally entirely because it was convenient, and prior to now nobody had carefully studied (a) whether the choice of ground metric matters or (b) how to implement WGANs with other ground metrics. In my opinion this paper is a solid contribution toward both. The theoretical contribution seemed sound to me, but I don't have a very extensive theory background so I defer to the other reviewers' opinions there. The experiments are generally carefully carried out and convincing. The authors achieve very competitive Inception scores with just a faithful (and careful, judging by the description in the supplementary) reimplementation of the WGAN-GP CIFAR-10 model, plus the method in the paper. An important caveat is that Inception scores are known to vary substantially between different random seeds of the same model hyperparameters [3], but the authors do re-train the model several times from scratch and plot the distribution in Fig 2. The paper could use additional experiments on different datasets and architectures. In particular I'd be very interested in a 128x128 image synthesis result, since it's plausible that the choice of norm is even more important in higher dimensions. Regarding figure 1, I'm not totally convinced by the visual assessment of samples or Fréchet derivatives. In particular, it's hard to tell whether the derivatives for s=-2 are have relatively greater low-frequency components than s=2, or whether the magnitudes for s=2 are just lower overall. Were the gradients rescaled to the same interval before visualization? The paper relies heavily on Inception scores for evaluation. This is understandable (they're a de-facto standard) but somewhat fraught: the experiments in this paper amount to evaluating different objectives (the BWGAN losses induced by different metrics) on the basis of their correlation with another objective (Inception score), which seems risky when the evaluation'' objective is heuristic. Unlike most other GAN papers, which argue (explicitly or implicitly) that improved scores result from more stable training dynamics, here the improvements are attributed mostly to the perceptual alignment of the loss. As such, it would be good to see at least one other evaluation metric reported; I suggest also reporting FID [1] scores. As a side result, the authors derive a heuristic for choosing hyperparameters which applies to both their method and WGAN-GP. Their baseline WGAN-GP outperforms the one in the original paper (which was state of the art at the time). It would be ideal to validate the heuristic empirically, for examply by sweeping over hyperparameters and showing that the heuristic achieves scores close to the optimal found by the sweep. In particular, in order for Figure 2 to support the conclusion that the choice of s matters, the other hyperparameters must be set to optimal or near-optimal values, which might themselves vary with s. The result (in the supplemental) that performance varies substantially between different L^p norms is significant and I feel should be included in the main paper, especially given that they basically matched the Sobolev norms in Inception score. Seeing that result changed my conclusion that Sobolev norms are an especially good choice for image modeling. The writing is generally clear and thorough. The math is easy to follow and the experiments are described in enough detail that reimplementation shouldn't be hard. It wasn't obvious before fully reading the text that the computation of the Fréchet derivative could be done implicitly'' by the trick on L172; the authors might want to highlight this in an algorithm box somewhere. The authors do a good job of providing background and placing their work within a larger context. Overall this paper explores an important topic, presents sound theory and a simple method, shows convincing (if slightly limited) experimental results, and is written clearly. Therefore I recommend acceptance.

Reviewer 2



======update===== While I do agree with reviewer 1 that this paper address the important problem of using alternative metrics other than L2 in WGAN-GP, I still believe the author(s) need to conduct more comprehensive set of experiments (not necessarily SOTA results as also mentioned by Reviewer #1). I have no doubt that this will be a very nice work, but not in its current form. It still feels bit of light to be presented at top notch conference venues such as NIPS, especially when compared to the gravity of its predecessors such as the original WGAN paper and the WGAN-GP paper. Their theories are synthesized from existing literature, and should have done a much better job clarifying the details to the generative modeling community. I particularly do not want to encourage publications with light experiments, which will set bad precedence. I would encourage the author(s) to further polish their work before resubmitting. It's a good one, but not ready yet. More specifically, 1) the only experiment used Inception Score can be very misleading (I am glad the author(s) include FID score in the rebuttal, but still not enough). The author(s) will also need more tractable toy examples to show the correctness of their heuristic automatic parameter setting procedure and the convergence of their algorithm to the ground truth distributions. 2) The stability of the algorithm and its computational efficiency is also aspects of practical concern, since forward and inverse FFT have been used which involve complex arithmetic. And how does the finite discretization estimator of the norm compare with the FFT estimator? 3) The image spaces X, the (image) function f and the coordinate system (also denoted as x in the paper) are heavily overloaded and it is very confusing. ================= This paper introduces Banach Wasserstein GAN which is a improved version of WGAN. The whole paper is easy to understand. Although this paper claim to reach the almost state-of-the-art inception score on CIFAR-10, the authors fail to compare their model using recent FID score. The authors also only test their model on just one dataset CIFAR-10, at least three different dataset need to be tested, i.e., MNIST, CelebA, Imagenet, LSUN. I also checked the supplementary, there is no other experiments. The authors also did not discuss the implementation details in both paper and supplementary material. Therefore, the whole paper is not convincing enough to me.

Reviewer 3



This submission deals extending the WGAN formulation with gradient penalty from Euclidean spaces to general bench spaces. To this end, the Sobolev norm and its dual are adopted into the earth mover distance, and the resulting cost function is shown to be the classic WGAN loss with a gradient penalty using the dual norm. The dual norm is argued to be computationally tractable for range of Banach norms, and experiments are provided with CIFAR data that show high inception score for some norms chosen from the Banach norms compared with the classic l2 norm. This is an interesting extension. The computational tractability and complexity of the gradient penalty calculation however is not very clear. More elaborations are needed on that side for various values of s. In addition, the inception score is not necessarily a reasonable metric to adopt, and readers rating are more reasonable to evaluate the quality of the resulting images. Thus, the experiment parts need more depth. Reviewer's opinion about the author response: My original concern was about relying on the inception score as the evaluation metric, where the rebuttal provides FID scores as well. This is appreciated and acceptable even though it still doesn’t truly reflect the readers subjective opinion score that is very common for instance in assessing the quality of outcomes for image superresolution algorithms. However, I still think the experiment section is not deep enough to prove the merits and usefulness of this new class of GANs. I think, to be coherent with the tone of this submission as a generalized class of WGANs, more extensive evaluations need to be reported for a few diverse and standard datasets such as ImageNet. Alternatively, the submission needs to be toned down which then does not merit publication at NIPS. Thus, I would recommend the authors to deepen the experimental results, and then this paper is of sufficient evidence to prove its merits.